# Preclinical Head and Neck Squamous Cell Carcinoma Models for Combined Targeted Therapy Approaches

**DOI:** 10.3390/cancers14102484

**Published:** 2022-05-18

**Authors:** Nina Schoenwaelder, Mareike Krause, Thomas Freitag, Björn Schneider, Sarah Zonnur, Annette Zimpfer, Anne Sophie Becker, Inken Salewski, Daniel Fabian Strüder, Heiko Lemcke, Christina Grosse-Thie, Christian Junghanss, Claudia Maletzki

**Affiliations:** 1Hematology, Oncology, Palliative Medicine, Department of Medicine, Clinic III, Rostock University Medical Center, 18057 Rostock, Germany; nina.schoenwaelder@med.uni-rostock.de (N.S.); mareike.krause@med.uni-rostock.de (M.K.); thomas.freitag@uni-rostock.de (T.F.); inken.salewski@med.uni-rostock.de (I.S.); christina.grosse-thie@med.uni-rostock.de (C.G.-T.); christian.junghanss@med.uni-rostock.de (C.J.); 2Institute of Pathology, Rostock University Medical Centre, 18057 Rostock, Germany; bjoern.schneider@med.uni-rostock.de (B.S.); sarah.zonnur@med.uni-rostock.de (S.Z.); annette.zimpfer@med.uni-rostock.de (A.Z.); anne-sophie.becker@med.uni-rostock.de (A.S.B.); 3Head and Neck Surgery “Otto Koerner”, Department of Otorhinolaryngology, Rostock University Medical Centre, 18057 Rostock, Germany; daniel.strueder@med.uni-rostock.de; 4Department of Cardiac Surgery, Reference and Translation Center for Cardiac Stem Cell Therapy (RTC), Rostock University Medical Center, University of Rostock, 18057 Rostock, Germany; heiko.lemcke@med.uni-rostock.de; 5Faculty of Interdisciplinary Research, Department Life, Light & Matter, University Rostock, 18057 Rostock, Germany

**Keywords:** cisplatin sensitivity, migration, invasion, phenotype, molecular alterations, preclinical tumor models, combined targeted approaches

## Abstract

**Simple Summary:**

Head and neck squamous cell carcinoma are characterized by a high degree of inter- and intratumoral heterogeneity. Well-characterized preclinical models represent the heterogeneity of this disease and enable the development of innovative therapeutic concepts. The present work dealt with the establishment of patient-individual tumor models to explore new treatment approaches for HNSCC patients and to identify suitable biomarkers for predicting treatment response. In this study, tumor specimens from advanced cancers of the oral cavity, hypopharynx, and larynx were used to establish individual tumor models. These novel cell lines were used to apply different combinations of strategies, preclinically, and to overcome intrinsic resistance mechanisms.

**Abstract:**

This study aimed to refine combined targeted approaches on well-characterized, low-passage tumor models. Upon in vivo xenografting in immunodeficient mice, three cell lines from locally advanced or metastatic HNSCC were established. Following quality control and basic characterization, drug response was examined after therapy with 5-FU, Cisplatin, and cyclin-dependent kinase inhibitors (abemaciclib, THZ1). Our cell lines showed different in vitro growth kinetics, morphology, invasive potential, and radiosensitivity. All cell lines were sensitive to 5-FU, Cisplatin, and THZ1. One cell line (HNSCC48 P0 M1) was sensitive to abemaciclib. Here, Cyto-FISH revealed a partial *CDKN2a* deletion, which resulted from a R58* mutation. Moreover, this cell line demonstrated chromosome 12 polysomy, accompanied by an increase in CDK4-specific copy numbers. In HNSCC16 P1 M1, we likewise identified polysomy-associated *CDK4*-gains. Although not sensitive to abemaciclib per se, the cell line showed a G1-arrest, an increased number of acidic organelles, and a swollen structure. Notably, intrinsic resistance was conquered by Cisplatin because of *cMYC* and *IDO-1* downregulation. Additionally, this Cisplatin-CDKI combination induced HLA-ABC and PD-L1 upregulation, which may enhance immunogenicity. Performing functional and molecular analysis on patient-individual HNSCC-models, we identified *CDK4*-gains as a biomarker for abemaciclib response prediction and describe an approach to conquer intrinsic CDKI resistance.

## 1. Introduction

Representative preclinical models are important tools to improve knowledge on tumor biology and treatment response. An ideal model preserves the phenotype and molecular features of the tumor, but also the complex and heterogeneous tumor microenvironment (TME) including immune and stromal cells. However, no available model covers all these characteristics [1]. Three experimental systems are currently used: long-term cultured cell lines, 2D- and 3D-patient-derived cultures, and xenograft models [1,2]. Each of them has specific advantages and limitations.

More than 800,000 patients per year are diagnosed with head and neck squamous cell cancer (HNSCC) worldwide. The mortality rate is almost 50% and the survivors suffer from massive functional limitations [3]. These tumors differ in location (e.g., oral cavity, pharynx, larynx), pathogenesis (human papillomavirus (HPV)-related vs. unrelated), mutational profiles (e.g., *TP53*-mutated and wild type), immune landscapes (immunologically “hot” vs. “cold”), and clinical prognosis (poor to moderate) [3,4,5,6]. The urgent need for biomarkers and effective personalized therapies demands well-characterized models that represent HNSCC’s heterogeneity to translate data from bench-to-bedside. Therefore, low-passage cell lines enable the studying of the intratumoral heterogeneity and clonal evolution under therapy pressure [7,8]. While surgery and radiotherapy remain standard in primary therapy, recurrent and metastatic HNSCC are treated with Cisplatin-based (radio-) chemotherapy or immune checkpoint inhibition to restore immune surveillance [9,10]. In HNSCC, implementing immunotherapy has only led to a slight increase in response rates and survival. The major reasons are insufficient biomarkers and resistance development. Preclinical tumor models assist in identifying more effective agents for HNSCC treatment. However, representative preclinical models are rarely available in this tumor entity. Moreover, recurrent and metastatic HNSCC are often treated without surgery and these tumors do not become available for preclinical research. This limitation is a hindrance for testing innovative treatment regimens. Still, there are some agents currently under investigation in early phase clinical trials. Cyclin-dependent kinase inhibitors (CDKIs) are a potential new HNSCC therapy. These agents were specifically designed to interfere with the tumors’ cell cycle [11,12,13]. Three CDK4/6 inhibitors are FDA-approved: palbociclib, ribociclib, and abemaciclib [14]. This latter agent is being used in several clinical trials for HNSCC treatment, either alone or in combination with anti-PD-1 antibodies (clinical trials.gov identifier: NCT03356223, NCT03356587, NCT03292250 NCT04169074, and NCT03356223). To increase the number of preclinical and patient-individual tumor models, we focused here on HNSCC from locally advanced or metastatic tumors. These novel models represent ideal tools for performing molecular analysis and drug screening. By using three individual models from different locations, we identified *CDK4* gains as a biomarker for abemaciclib response. These novel patient-individual HNSCC models represent the diseases’ heterogeneity and help to guide combination strategies for upcoming clinical application.

## 2. Materials and Methods

### 2.1. Tumor Sample Preparation

HNSCC samples were received fresh from surgery at the University Medical Center as described [15]. Written informed consent from the HNSCC patients was obtained according to the local Ethics Committee (reference number A2018-0003) and the guidelines for the use of human material (Declaration of Helsinki).

### 2.2. Ethical Statement

#### 2.2.1. In Vivo Study

The German local authority approved all animal experiments: Landesamt für Landwirtschaft, Lebensmittelsicherheit und Fischerei Mecklenburg-Vorpommern (7221.3-1-066/18 and 7221.3-1-032/19-4), under the German animal protection law and the EU Guideline 2010/63/EU. Mice were bred in the animal facility of the University Medical Center in Rostock under specific pathogen-free conditions. All animals received enrichment as mouse-igloos (ANT Tierhaltungsbedarf, Buxtehude, Germany), nesting material (shredded tissue paper, Verbandmittel GmbH, Frankenberg, Deutschland), paper roles (75 × 38 mm, H 0528–151, ssniff-Spezialdiäten GmbH), and wooden sticks (40 × 16 × 10 mm, Abedd, Vienna, Austria). During the experiment, mice were kept in type III cages (Zoonlab GmbH, Castrop-Rauxel, Germany) at a 12-h dark:light cycle, a temperature of 21 ± 2 °C, and relative humidity of 60 ± 20% with food (pellets, 10 mm, ssniff-Spezialdiäten GmbH, Soest, Germany) and tap water ad libitum.

#### 2.2.2. Patient-Derived Xenograft (PDX) Generation

Six-week-old female NOD.Cg-Prkdc^scid^Il2rg^tm1^Wjl (NSG, Charles River Laboratories, Lyon, France) mice were used as recipients. A detailed protocol is described in [15]. The patients’ serial numbers were maintained throughout all PDX passages. Cell culture was started after the first or second in vivo passage.

#### 2.2.3. Experimental Protocol

Cell-line derived xenografts were generated by injecting 5 × 10^6^ cells of HNSCC16 P1 M1, HNSCC46 P0 M2, and HNSCC48 P0 M2 subcutaneously in the right flank of 6–8 weeks old female NSG mice. Tumor diameters were measured with a caliper every three to four days. Tumor volumes were calculated as (length × width^2^)/2. Mice were euthanized before tumors reached 1.5 cm³. An in vivo therapy approach was carried out with HNSCC16 P1 M1. If the mice were bearing tumors of ~50 mm^3^, they were allocated to the control or the treatment group.

### 2.3. Collagenase Digestion for Ex Vivo Primary Cell Cultures

Collagenase digestion was carried out with minor modifications according to [16]. Vital tumor tissues were cut into small pieces of approximately 2 mm³ and washed two times with phosphate-buffered saline (PBS). Collagenase (2.5 mg/mL, Nordmark Biochemicals, Uetersen, Germany) was dissolved in 0.5 M tris(hydroxymethyl)aminomethane and 150 mM Calcium chloride. Collagenase digestion was carried out for 1.5–2 h at 37 °C with regular vortexing. Cells were washed with PBS and suspended in medium: DMEM/Hams F12 supplemented with 20% fetal calf serum (FCS), glutamine (2 mmol/L), and antibiotics (medium and antibiotics were from Pan Biotech, Aidenbach, Germany, FCS from Sigma–Aldrich, Darmstadt, Germany and glutamine from Biochrom, Berlin, Germany). Single cells were obtained by using a 100 µm cell strainer. Single cells and remaining tissue were separated into two different 6-wells. Medium was changed regularly. Growing cell cultures were seeded into 25 cm^2^ flasks and the serum concentration was serially reduced to 10% (with exception of HNSCC48 P0 M1: final serum concentration 15%).

### 2.4. Spheroid Formation

A total of 5000 cells were seeded in serum free DMEM/Hams F12 supplemented with 1× NCS21 (50×; Capricorn Scientific, Ebsdorfergrund, Germany), 20 ng/mL recombinant human epidermal growth factor (rhEGF), and 10 ng/mL recombinant human fibroblast growth factor (rhFGF-b) (both from ImmunoTools, Friesoythe, Germany) in ultra-low-attachment (ULA) plates (Greiner Bio-One, Kremsmünster, Austria). Spheroids formed after ~72 h.

### 2.5. Mycoplasma and Fibroblast Contamination Test

The MycoAlert Mycoplasma Detection Kit (LONZA, Rockland, ME, USA) was used to exclude a mycoplasma contamination. Analysis was carried out following the manufacturer’s instruction. To exclude fibroblast contamination, cells were stained with anti-CD90 and anti-CD326 antibodies (both 1:50, BioLegend, San Diego, CA, USA) for 30 min at 4 °C followed by washing and analysis on a FACSVerse Cytometer (BD Pharmingen, Heidelberg, Germany). Data analysis was performed using BD FACSuite software (BD Pharmingen).

### 2.6. Histology and Immunohistochemistry

The morphology of the patients’ tumor tissue and their corresponding PDX was studied by an expert pathologist. Histopathology of primary tumors and PDX followed standard protocols for HNSCC diagnostics including HE staining and immunohistochemistry; antibodies: anti-p16 (clone: G175-405, BD Bioscience, Heidelberg, Germany), anti-Ki-67 (Clone Mib-1, Dako, Glostrup, Denmark), anti-PD-L1 (Clone 22C3, Dako), and anti-p53 (Clone Do7, Dako). Standard immunoperoxidase technique was applied using an automated immunostainer, Autostainer Link48 (DAKO) with diaminobenzidine as chromogen.

### 2.7. Tumor Microenvironment

A single cell suspension of the patients’ tumor was stained with 1 µg of each of the following surface markers: anti-CD3 FITC (clone OKT-3), anti-CD4 PE (clone IT4), anti-CD8 PE (clone MEM-31), anti-CD56 PE (clone MEM-188), anti-CD19 APC (clone LT19), anti-CD14 FITC (clone 63D3), anti-CD163 PECy7 (clone GHI/61), anti-CD169 APC (clone 7-239), anti-CD204 PE (clone 7C9C20), anti-CD274 PECy7 (clone 29E.2A3), anti-CD276 PE (clone DCN.70), anti-CD273 APC (clone 24F.10C12). Antibodies were incubated for 30 min at 4 °C. Finally, cells were washed, suspended in PBS, and analyzed with FACSVerse Cytometer (BD Pharmingen). Data analysis was performed using BD FACSuite software (BD Pharmingen).

### 2.8. Molecular Pathology

HPV status and genomic alterations were analyzed using the Illumina Cancer Hotspot Panel (Illumina, Berlin, Germany) as described previously [15]. As a quality control, a DNA Fingerprint PCR of matched tumor tissue, PDX, and cell line was carried out as described [17]. To check for gene specific amplifications 30,000 cells per cell line were fixed on a coated cytoslide (THARMAC cellspin, Wiesbaden, Germany) by using the SHANDON cytospin3 centrifuge cell preparation system. Centrifugation was conducted for 10 min at 700 rpm. After 24 h, the gene specific staining of *CDKN2A* and *CDK4* was conducted according to the manufacturer’s instructions. The staining kits SPEC CDK4/CEN 12 Dual Color Probe and SPEC CDKN2A/CEN9 Dual Color Probe (ZytoVision, Bremerhaven, Germany) were used.

### 2.9. In Vitro Growth Kinetics

Population doubling times (PDT) were determined by seeding 500,000 cells into replicate 25 cm^2^ (HNSCC46 P0 M2, HNSCC48 P0 M1) or 75 cm^2^ flasks (HNSCC16 P1 M1) and counted daily for five days. The doubling time was calculated as follows: PDT = t  ×  ln2/ln(N_x_/N_0_), t = incubation time between N_x_ and N_0_, N_x_ = total amount of cells at the end of the exponential growth and N_0_ = total amount of cells at the beginning of the exponential growth.

### 2.10. Phenotyping

For initial phenotyping, cells were left untreated and stained for 30 min at 4 °C with FITC anti-HLA-ABC antibody (MHC I) (1:50; both from ImmunoTools), PE/Cy7 anti-CD274 (PD-L1), and APC anti-CD279 (PD-1) (1:50; both from BioLegend; blue (488 nm) and red (633 nm) laser). The antibodies were diluted in 2% BSA with 2 mM Ethylenediaminetetraacetic acid disodium salt solution (EDTA) (Sigma–Aldrich, Darmstadt, Germany). Finally, cells were analyzed with FACSVerse Cytometer (BD Pharmingen). Data analysis was performed using BD FACSuite software (BD Pharmingen). Additionally, untreated cells were stained with anti-CD279 (PD-1) (1:50) at 4 °C overnight. Cells were washed and cell nuclei were stained with 4′,6-diamidino-2-phenylindole (DAPI) (AAT Bioquest, Sunnyvale, CA, USA). After, a second washing step analysis was performed on a ZEISS Elyra 7 Confocal Laser Microscope (Zeiss, Jena, Germany). To investigate the effect of the applied test substances on the cell surface markers HLA-ABC and PD-L1, the cells were treated for 2 × 72 h and stained as described for the initial treatment.

### 2.11. Radiation Response

A total of 5000 cells (HNSCC16 P1 M1, HNSCC48 P0 M1) or 15,000 cells (HNSCC46 P0 M2) were seeded in 100 µL in three technical replicates/cell line in 96-well plates. Irradiation was started after four days with 2 Gy and 14 Gy single radiation dose (Cs-137 γ-irradiation; IBL 637, CIS Bio-International, Codolet, France). Cells were irradiated for five days daily. On the third day of irradiation, 100 µL fresh medium was added. Three days after the last irradiation, a crystal violet assay was carried out.

### 2.12. Migration and Invasiveness

Migration and invasion was examined as described [18].

### 2.13. In Vitro Drug Response

Cells were seeded in 96-well plates in three technical replicates and treated after 24 h incubation at 37 °C and 5% CO_2_. Cells were treated for 2 × 72 h in monotherapy with the different test substances in varying concentrations ranging from 0.048 µg/mL to 1 mg/mL for approved drugs (Cisplatin, 5-Fluorouracil (5-FU), Cetuximab, pharmacy of the University Hospital Rostock) and 1 nM–1 µM for targeted substances (abemaciclib, dinaciclib, Selleckchem, Munich, Germany, THZ1 Hycultec, Beutelsbach, Germany). The level of 50% inhibition (IC_50_) was calculated using the IC_50_ calculator from AAT Bioquest (https://www.aatbio.com/tools/ic50-calculator/ (accessed on 1 November 2021)). A combination was carried out in a simultaneous setting for 2 × 72 h hours using the following concentrations: abemaciclib [HNSCC16 P1 M1/HNSCC46 P0 M2: 1 µM, HNSCC48 P0 M1: 0.1 µM], dinaciclib [HNSCC16 P1 M1/HNSCC46 P0 M2: 1 nM, HNSCC48 P0 M1: 5 nM], THZ1 [HNSCC16 P1 M1/HNSCC48 P0 M1: 20 nM, HNSCC46 P0 M2: 10 nM], and Cisplatin [HNSCC16 P1 M1/HNSCC46 P0 M2: 1 µg/mL, HNSCC48 P0 M1: 0.05 µg/mL]. A readout was conducted with crystal violet staining. The Bliss Independence model was used to analyze potential synergistic and additive effects between the substances.

### 2.14. Cell Death and Cell Cycle

An apoptosis–necrosis assay was carried out after 2 × 72 h of treatment, supplemented by cell cycle analysis after 2 × 72 h treatment as described previously [18].

### 2.15. Assessment of Viability, Acidic Compartments, Cytoskeleton and ROS

Acidic compartments were visualized as described [19]. Therefore, cells were treated for 2 × 72 h. The influence on cytoskeleton and formation of reactive oxygen species (ROS) were likewise analyzed after 2 × 72 h treatment. Cells were washed with HHBS and stained with ROS Brite 670 (7.5 µM, AAT Bioquest) for 30 min at 37 °C. Subsequently, cells were permeabilized, fixed, and stained with Phalloidin-iFluor 594 conjugate (1:1000, AAT Bioquest) for 30 min at room temperature. Cells were washed and cell nuclei stained with DAPI. After a second washing step analysis was performed on a ZEISS Elyra 7 Confocal Laser Microscope (Zeiss, Jena, Germany).

### 2.16. RNA Isolation, cDNA Synthesis, and Quantitative Real-Time PCR

RNA Isolation, cDNA Synthesis, and Quantitative Real-Time PCR was carried out as described [20]. Reactions were performed in triplicates and repeated three times. *GAPDH* was self-designed and used as a housekeeping gene (GAPDH forward: TCACCAGGGCTGCTTTTAAC; GAPDH reverse: GGGTGGAATCATATTGGAACA; GAPDH 5′ HEX-3′ BHQ-1 TGCCATCAATGACCCCTTCATTG). *MYC* (Hs00153408_m1), *PDK2* (Hs00176865_m1), *SKP2* (Hs01021864_m1), and *IDO-1* (Hs00984148_m1) were 6-FAM labeled and purchased from ThermoFisher Scientific (Waltham, MA, USA).

### 2.17. Allogenic Co-Culture

HNSCC48 P0 M1 was stained with 5 µM 5-(and-6)-Carboxyfluorescein diacetate, succinimidyl ester (CMFDA; Biotium, Fremont, California, USA) before seeding. 24 h later, PBMCs (Peripheral Blood Mononuclear Cells) were isolated by a density gradient centrifugation and added (tumor cell:PBMC ratio: 1:10). PBMCs were stimulated with 100 IU/mL of IL-2 (Novartis, Basel, Switzerland). Abemaciclib [0.1 µM], Pembrolizumab [10 µg/mL] or the combination of both were added simultaneously with the PBMCs. After 1 × 72 h, cells were harvested and collected in FACS tubes. For the readout, 50 µL of 1:35 diluted beads (fluorescent microsphere beads; 1.4 × 10^4^ beads/mL; size 10 µm; Polyscience, Warrington, PE, USA) were added to the 200 µL of cell suspension in the respective FACS tubes. Measurement of the beads and residual tumor cells stopped when 5000 beads were counted.

### 2.18. Statistics

All values are given as mean ± SD (in vitro analysis) or mean ± SEM (in vivo approach). Statistical evaluation was performed using GraphPad PRISM software, version 8.0.2 (GraphPad Software, San Diego, CA, USA). The criterion for significance was set to *p* < 0.05. After proving the assumption of normality (Shapiro–Wilk test), one-way ANOVA (Dunnett’s multiple comparison), two-way ANOVA (Tukey’s multiple comparison) or T-test was performed. If the normality test failed, the Kruskal–Wallis or U-Test was performed.

## 3. Results

### 3.1. Clinicopathological Patients’ Data, and Cell Line Establishment

Three novel HNSCC cell lines HNSCC16 M1 P1, HNSCC46 P0 M2, and HNSCC48 P0 M1 were generated after xenografting in NSG mice. Clinicopathological patients’ characteristics are summarized in Table 1.

### 3.2. Histopathology and Tumor Microenvironment of Primary Tumors

The histological pattern of all three cases is that of a moderately differentiated squamous cell carcinoma (Figure 1A). HNSCC48 shows an expression pattern typical for *TP53* mutations; while in the other cases, the p53-expression pattern is not mutation-specific. Because of tissue processing artifacts, the immunohistochemistry data (Ki-67, p53, and PD-L1) for HNSCC46 are not fully exploitable.

The TME was heterogeneous (Figure 1B). The strongest lymphocytic infiltration was detectable in HNSCC16 with CD3^+^CD4^+^/CD3^+^CD8^+^ T cells and CD19^+^ B cells. CD16^+^CD56^+^ natural killer (NK) cells were found in all cases with the highest abundance in HNSCC46. This tumor case additionally showed the highest number of tumor-associated macrophages (TAM). The infiltrating CD3^+^ T cells showed a low level of CD274^+^ (no data available for HNSCC48). Roughly 15% of T cells were CD273^+^ (=PD-L2) and numbers comparable between the three HNSCC cases, while CD276^+^ was highest on HNSCC46/HNSCC48 (vs. HNSCC16).

### 3.3. Quality Control, Comparative Analysis of the Molecular Profile & Detection of Gene Specific Alteration of CDK4 and CDKN2A

Using fingerprint PCR as quality control, the ancestry of the cell lines with its corresponding PDX and patient’s tumor were validated (Appendix A). *TP53* hotspot mutations (single or multiple) were detectable in all cases (Table 2). The patient’s tumor HNSCC16 harbored an additional p.P27R mutation that was lost during passage. In HNSCC48, the *CDKN2A* mutation p.R58* and the *SMAD4* mutation p.R135* was found. The variant allele frequencies (VAF) of individual mutations revealed higher VAFs in the PDX and cell line compared to the patients’ tumors (Table 2). The only exception is HNSCC48.

Gene-specific amplifications of *CDKN2A* (located on 9p21) and *CDK4* were checked as potential biomarkers. All cell lines showed chromosome 9 polysomy with high copy numbers of the respective centromer (Figure 2). In HNSCC16 P1 M1/HNSCC46 P0 M2, *CDKN2A* gains were equal to chromosome 9 copies. In HNSCC48 P0 M1, fewer copies of the *CDKN2A* gene compared to the number of copies of the centromere of chromosome 9 were found (ratio: 3:1), indicative for a gene specific *CDKN2A* deletion and thus confirming sequencing data.

*CDK4* is of high relevance for CDKI treatment. HNSCC16 P1 M1/HNSCC48 P0 M1 demonstrated polysomy of the chromosome 12, with the latter having 4–6 chromosomes/cell (Figure 2). Notably, this was accompanied by an increase in gene-specific copy numbers and was most pronounced in HNSCC48 P0 M1.

### 3.4. Cell Morphology and Spheroid Formation

All cells reveal tight adherence to the flask bottom with different morphology (Appendix A). HNSCC48 P0 M1 changed morphologically during culture and a phenotypically small cell clone dominated at later passages. A fraction of HNSCC16 M1 P1 cells formed spheroidal-like cell clusters spontaneously, which were preserved under spheroid-forming conditions (Appendix A). Fibroblasts were excluded by flow cytometry (CD90/EpCAM (CD326) staining) (Appendix A).

### 3.5. Cell Line Characterization

The PDT differed among cells (Appendix A). Flow cytometric phenotyping revealed HLA-ABC and PD-L1 positivity on a fraction of cells (Figure 3A). For example, in HNSCC16 P1 M1, approximately one-third of cells was found to be positive for HLA-ABC and three-quarters were positive for PD-L1 (Figure 3A, left panel). Single cells of HNSCC46 P0 M2 were additionally PD-1^+^ (Figure 3A, right). Immunofluorescence confirmed PD-1 expression on a fraction of cells in this line, while others were completely negative (Figure 3B, middle). The radiation response was quite heterogeneous: HNSCC48 P0 M1 was most radiosensitive. Increasing the total radiation dose to 70 Gy induced a significant decline in all three cell lines (Figure 3B). Assessment of migration potential revealed fast scratch closure in two-thirds of cases (Figure 3C,D). The invasiveness was compared to UT-SCC-15, a cell line with known invasive behavior [18]. HNSCC16 P1 M1/HNSCC46 P0 M2 were more invasive than UT-SCC-15 and HNSCC48 P0 M1, confirming migration results.

### 3.6. In Vitro and In Vivo Drug Response

Drug response analysis was carried out after two rounds of treatment with each treatment cycle lasting 72 h. The dose–response curves are given in Appendix A and the resulting IC50 values are presented in Table 3.

All cell lines were sensitive towards Cisplatin and 5-FU, while an individual response profile was seen for Cetuximab and the CDKIs. On a basis of this initial screening, combination strategies were applied to check for potential synergistic effects. Therefore, the IC_20_ was applied, and functional analysis was carried out to decipher underlying molecular mechanisms. Using a combined chemo-CDKI approach synergistic effects were seen upon Cisplatin and abemaciclib/THZ1 in HNSCC16 P1 M1/HNSCC48 P0 M1 cells (Figure 4A). Vice versa, these combinations were antagonistic in HNSCC46 P0 M2, nicely reflecting the HNSCCs heterogeneity. For the latter case, all combinations failed to be synergistic (Figure 4A). In an in vivo trial, drug response against Cisplatin was examined and HNSCC16 M1 P1 was chosen as experimental model, due to the comparably fast and reliable in vivo growth kinetics (Figure 4B). As therapeutic agent, we applied Cisplatin because all cell lines showed high sensitivity in vitro. Also, this drug is still widely applied in the clinic for the treatment of HNSCC patients both in mono- or combination therapy. In this initial study, Cisplatin slightly decelerated growth; still, no significant growth reduction was seen (Figure 4C) and trials were stopped prematurely.

### 3.7. Molecular Alterations Predict CDKI Resistance, Which Is Partially Reversible by Combined Targeted Therapy In Vitro

The applied treatment schedules induced apoptosis and/or necrosis to varying degrees (Figure 5A). Notably, dual CDK inhibition (i.e., abemaciclib + THZ1) was comparably effective as Cisplatin-tailored CDKI combinations. Autophagy, as indicated by acidic organelles, was additionally seen after Cisplatin and abemaciclib mono- and combination therapy in HNSCC16 P1 M1 (Figure 5B). In HNSCC46 P0 M2, the effect was weaker but still visible. Interestingly, all monotherapies reduced the level of reactive oxygen species (ROS), whereas Cisplatin with THZ1 increased ROS in HNSCC16 P1 M1 (Figure 5C).

Thereafter, resistance-associated genes were examined, because of the promising effects of combined Cisplatin-CDKI application (Figure 6): basal expression levels confirmed an upregulation of *SKP2* and *cMYC*, but not *PDK2* or *IDO-1* compared to normal mucosa (Figure 6A). Cisplatin further induced *SKP2* in two-thirds of cases, while *SKP2* was counter regulated in the combination (Figure 6B). HNSCC46 P0 M2 and HNSCC48 P0 M1 responded with an upregulation of *cMYC* and *IDO-1* upon abemaciclib. In HNSCC46 P0 M2, upregulation was repealed by Cisplatin. *PDK2* was massively induced in HNSCC46 P0 M2 after Cisplatin and abemaciclib monotherapy, but the combination again revoked this effect.

As the cell cycle is the main target of CDKIs, this was analyzed by flow cytometry (Figure 6C). In HNSCC16 P1 M1 and HNSCC48 P0 M1, abemaciclib and THZ1 induced a G1 phase arrest, which was not enhanced in the combination. Cisplatin led to a G2 phase arrest that was preserved in combination with THZ1. In HNSCC46 P0 M2, effects were again weaker and mainly attributable to Cisplatin (Figure 6C). Adding CDKIs to Cisplatin slightly boosted effects of the monotherapy.

### 3.8. Enhanced Immunogenicity and Confirmation of Immune-Stimulatory Potential upon Co-Culture with Abemaciclib and Pembrolizumab

The abundance of HLA-ABC increased after abemaciclib monotherapy and in combination with Cisplatin or THZ1 in all three cases (Figure 7A). The highest increase was seen on HNSCC48 P0 M1, in which PD-L1 was additionally upregulated. HNSCC16 P1 M1 showed comparable, though less strong, phenotypic changes. No effect on PD-L1 was recognizable on HNSCC46 P0 M2 cells.

In a final in vivo-like co-culture system HNSCC48 P0 M1 cells were included because of their PD-L1-positivity and abemaciclib sensitivity (Figure 7B). Tumor and immune cells were co-cultured in the presence of abemaciclib, the anti-PD-1 inhibitor Pembrolizumab or a combination of both to test which strategy exerts highest immune-mediated tumor killing. Abemaciclib reduced cell viability to ~50% (vs. control). Pembrolizumab effectivity was donor specific. While residual cells decreased to 39% in Donor B and 83% in donor D PBMC co-culture, the amount of tumor cells remained unchanged for donor A and C. Interestingly, the combination of abemaciclib and Pembrolizumab was most effective and donor independent (donor A: 44%; donor B 39%; donor C 43%; donor D 26%).

## 4. Discussion

In this study, we established three novel xenograft-derived HNSCC cell lines to address intertumoral heterogeneity. We showed that the TME of the patients’ tumors consists of varying numbers of infiltrating lymphocytes, TAMs, and immune checkpoint molecules. All tumors were resected at advanced stages, allowing us to investigate therapies on low-passage models with particular relevance for patients with poor prognoses [4]. In fact, representative well-characterized preclinical HNSCC models are rarely available. This is mainly attributable to the fact that patients with recurrent and metastatic disease are usually treated with systemic therapy, without surgery. The paucity of such HNSCC models precludes preclinical research that can be translated into clinical trials. To fill this gap, novel tumor models from advanced HNSCCs are warranted. This is of particular clinical relevance, knowing that the currently available drugs have limited efficacy in many patients.

To address this issue, we initially tested HNSCC cell line establishment directly from patient tumor samples. Though this was described before [8,21], we failed to generate cell lines because of overgrowing fibroblasts (data not shown). We moved on with the NSG mouse PDX model to generate cell lines from PDX and to exclude fibroblast overgrowth [16]. Using this method, we generated three novel cell lines (and more are being established). All cells were defined as epithelial tumor cells (EpCAM^+^, CD90^−^) with a molecular profile reflecting HPV-unrelated HNSCC. This included *TP53* mutations in all cases [22], along with *SMAD4* and *CDKN2A* mutations in HNSCC48 [23]. *CDKN2A* cyto-FISH identified a partial gene deletion accompanied by chromosome 9 polysomy. In the other two cases, *CDKN2A* gains were equal to chromosome 9 copies (HNSCC16, n = 4–5/cell) or remained unchanged with only chromosome 9 polysomy (HNSCC46). Chromosome polysomy is a common characteristic of advanced tumors and was described in p53 dysregulated HNSCC before [24].

The direct comparison between patients’ tumors, PDX, and the corresponding cell line revealed preservation of the molecular profile—at least in early passages. This analysis is imperative to guarantee high-quality data generation and to correlate molecular features of the parental tumor with drug response. Rather unexpected were the VAFs of the PDX from HNSCC48. In HNSCC16 and HNSCC46, the VAFs remained stable or increased during transfer, likely because of increasing amounts of mutated cell clones and/or outgrowth of one dominating clone in culture. But in the HNSCC48 case, the VAF transiently decreased during passage, likely because of intratumoral heterogeneity. It can thus be assumed that a subclone in the tissue used for the PDX generation could not adapt to in vitro conditions. This phenomenon shows a limitation of our study and of cell lines in general and can only be vanquished by including different regions of one tumor. Still, this is laborious, time-consuming, and costly and thus hardly realizable. Solid cancers initially originate from one mutated cell clone with sustained proliferation ability and immune evasion capacity. But with disease progression, subclones develop and different genomic alterations occur, as a result of genomic instability. In HNSCC, up to six different subclones were found [6,25]. Hence, cell cultures can only partially reflect the in vivo situation and only about one-third of somatic mutations are ubiquitously detectable in every tumor region [25,26]. By collecting background information via NGS, TME, and comprehensive evaluation of the drug response, we aimed to overcome such limitations and provide helpful information for follow-up studies.

Our cell lines exhibited various growth kinetics, radiation sensitivity, migration/invasion potential, and only HNSCC46 P0 M1 was positive for PD-1. Two of three cell lines formed solid spheroids in vitro and engrafted rapidly in vivo. HNSCC46 P0 M2 is the only exception, forming loose spheroids and showing slow engraftment rates, thus reflecting the reduced population doubling time in vitro. Still, the fact that two-thirds of cell lines formed spheroids allows further research to be a step closer to the in vivo setting. Such 3D models recapitulate interactions in the TME, transport properties, oxygen, nutrient, and proliferation gradients [27].

Radiosensitivity was classified based on clinical doses [28]. All cell lines were sensitive to cytostatics (5-FU, Cisplatin) and global-acting CDKIs, however, with individual differences. HNSCC48 P0 M1 was additionally vulnerable to abemaciclib and Cetuximab. Cytotoxic effects matched with those described in other entities [29,30]. Whether such cross-sensitivity/-resistance is a unique finding or a general phenomenon, prospective examination on a larger cell line panel is necessary. Notably, however, is the fact that abemaciclib resistance was reversible in HNSCC16 P1 M1 cells by adding Cisplatin. Regarding HNSCC48 P0 M1, we conclude the partial deletion of *CDKN2A* due to a pathogenic R58* mutation may be the most likely molecular correlate for abemaciclib response [31,32]. Vice versa, HNSCC16 P1 M1 harbored wild type *CDKN2A*. We therefore checked for *CDK4* gene amplifications via cyto-FISH because abemaciclib has a higher selectivity for CDK4 than for CDK6 [33]. HNSCC48 P0 M1 cells displayed the expected chromosome 12 polysomy along with *CDK4* gains. Such polysomy-associated *CDK4* gains were also detectable in HNSCC16 P1 M1, but not HNSCC46 P0 M2 cells, which may explain our findings best. This is supported by the observation of morphological changes, such as an increased number of acidic organelles, a swollen cell structure and also the G1 phase arrest upon abemaciclib monotherapy in HNSCC16 P1 M1. We therefore propose *CDK4* gains as a biomarker for response to abemaciclib, additionally to the known *CDKN2A* deletion. This allows us to conclude that polysomy and the associated increased copy number of the *CDK4* gene increases the abemaciclib response (especially when combined with Cisplatin). HNSCC46 P0 M2 was the most chemotherapy resistant and “special”. Though, eventually coincidentally, it is worth mentioning that the TME of the patients’ tumors harbored a high number of TAMs and NK cells as likely indicators for poor drug response. Such a link between the TME and cellular characteristics has been described for (intrinsic drug-resistant) cancer-stem cells only [34] and demands further investigation.

The challenge to improve the drug response is partially attributed to the upregulation of genes mediating drug resistance. *CMYC* is a prominent marker and was also higher in our cell lines than in adjacent normal mucosa [35]. Abemaciclib-induced upregulation was partially counter regulated by Cisplatin. A previous study reported contrary effects in thyroid cancer, however, at 10–20-fold higher abemaciclib doses [36]. We interpret our finding as an acute stress response, underscored by an upregulation of *IDO-1* after abemaciclib treatment. Future research should address sequential pharmacological inhibition of IDO-1 (Indoximod or Epacadostat) [37] to boost CDKI efficacy. Noteworthy from an immunological point is the upregulation of HLA-ABC/PD-L1 by abemaciclib and its combinations, likely because of CDK4 inhibition. CyclinD-CDK4 is involved in mediating PD-L1 ubiquitination and degradation [38,39]. Given the fact that immune-checkpoint inhibitors are approved as first-line standard treatment for recurrent and metastatic HNSCC patients, abemaciclib may provide a perspective for the majority of patients that do not respond to PD-1 antibodies. Indeed, a very recent study on patients with advanced, refractory solid tumors described clinical benefits in CDKI/immune-checkpoint inhibitor combinations vs. the respective monotherapy [40]. Using an allogenic co-culture system, we demonstrated promising results upon combined abemaciclib and Pembrolizumab therapy, which even exceeded the effects of the monotherapy. However, this must be further investigated in an autologous setting using “tumor-edited” T cells. In our HNSCC biobank [15], we routinely collect autologous lymphocytes along with tumor samples. Hence, performing such sophisticated analyses on primary tumors and immune cells will help in becoming a step closer to “in vitro immunotherapy”.

Summing up our findings, these newly established patient-individual tumor models guide the selection of drugs (combinations) and help in detecting intrinsic or acquired resistance mechanisms. With our complex set of analyses, we not only addressed morphological, molecular, and immunological aspects of advanced HNSCC, but also identified *CDK4* gains as a new “surrogate” marker for abemaciclib response and describe a personalized approach to conquer intrinsic CDKI resistance.

## 5. Conclusions

This work shows that well-characterized patient-individual tumor models are crucial for identification of effective therapy (combinations) and play an important role in detecting intrinsic resistance mechanisms. However, it also illustrates that in light of the intertumoral heterogeneity of HNSCC, a comprehensive characterization is necessary to draw general conclusions between the cell lines investigated or to explain different results and to identify potential new biomarkers.

Here, we propose CDK4 gains as a biomarker for response to abemaciclib. Of note, abemaciclib resistance was reversible in one case (HNSCC16 P1 M1) by Cisplatin. This highlights the potential for combination therapies, especially in the context of intrinsic resistance towards targeted therapies.

This study provides further evidence for the therapeutic potential of CDKIs and shows that they can be combined with classical chemotherapy. These findings contribute to our understanding of how the treatment of HNSCC can be improved in the future.

## Figures and Tables

**Figure 1 cancers-14-02484-f001:**
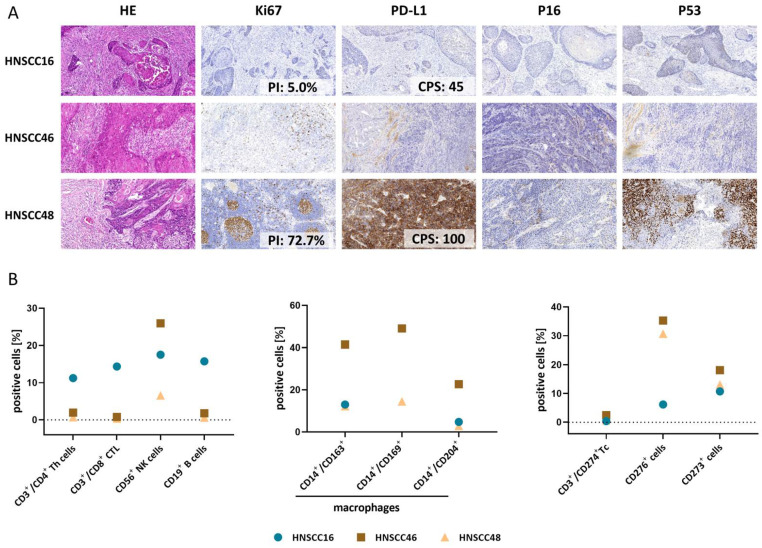
Histopathology and tumor microenvironment of primary tumors. (**A**) HE-, Ki67-, PD-L1, P16-, and P53-staining of the patients’ tumors HNSCC16, HNSCC46, and HNSCC48. 10 × magnification. (**B**) Tumor microenvironment of the primary tumors HNSCC16 (turquoise circles), HNSCC46 (brown squares), and HNSCC48 (orange triangles). Relative fractions of lymphocytes (left diagram), TAMs (middle diagram), and selected checkpoints (right diagram) are illustrated.

**Figure 2 cancers-14-02484-f002:**
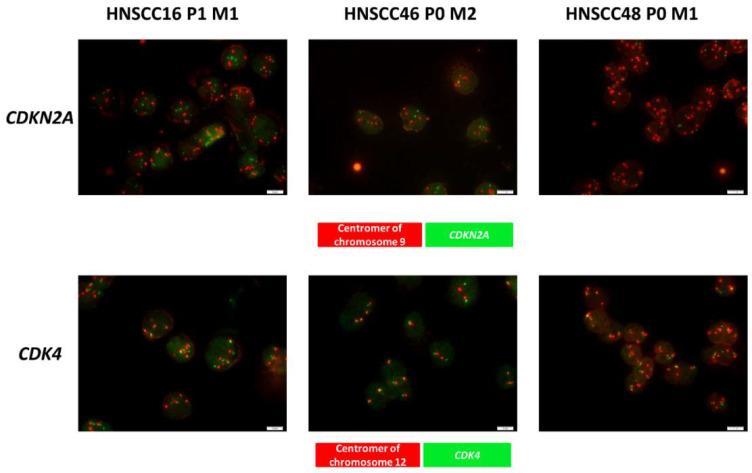
Cyto-FISH for *CDKN2a* and *CDK4*. Cytospins of HNSCC cell lines were stained with the SPEC CDK4/CEN 12 Dual Color Probe or SPEC CDKN2A /CEN9 Dual Color Probe to check for gene specific amplification of *CDKN2A* and *CDK4*. The red spots indicate the centromers and the green spots indicate the specific gene. A readout was carried out with the fluorescence microscope Olympus BX53. Original magnification 1000×.

**Figure 3 cancers-14-02484-f003:**
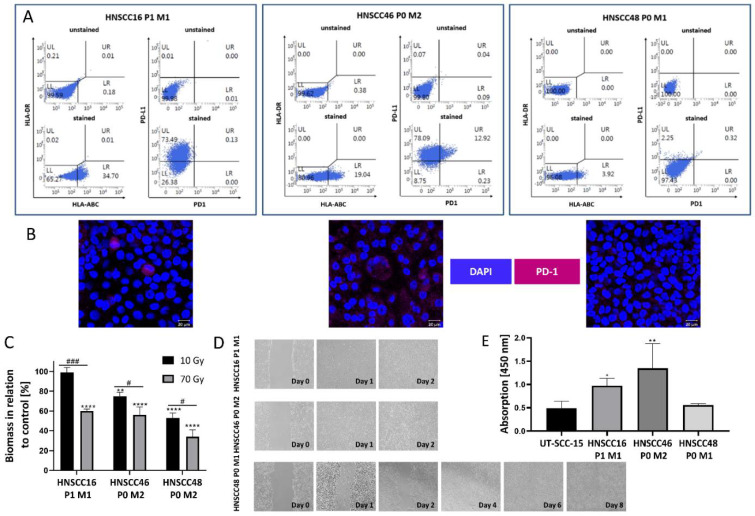
Basal characterization including phenotyping, radiation response, migration, and invasion of untreated HNSCC cell lines (**A**) FACS-based phenotyping for HLA-ABC, PD-1, and PD-L1 (left) and (**B**) immunofluorescence staining of PD-1 (right) to confirm PD-1 on the surface of HNSCC46 P0 M2. A read out was conducted with the ZEISS Elyra 7 Confocal Laser Microscope (Zeiss). Original magnification 400×. (**C**) Radiation response was examined after 5 fractions of 2 Gy and 14 Gy on 5 consecutive days, resulting in a total radiation dose of 10 Gy (black columns) and 70 Gy (grey columns). A readout was conducted by crystal violet staining and biomass quantified in relation to non-irradiated controls. Unpaired t-test (n = 3 independent experiments) ^#^
*p* < 0.05; ^###^
*p* < 0.001 vs. 70 Gy; one-way ANOVA, (n = 3 independent experiments) ** *p* < 0.01; **** *p* < 0.0001 vs. control (100%). (**D**) Migratory potential was investigated via scratch-assay. The scratch was monitored daily by light microscopy (original magnification 100×) until the wound was closed. (**E**) Matrigel-invasion assay. Invasive behavior was studied compared to UT-SCC-15, a cell line derived from a nodal HNSCC recurrence. A readout was conducted with WST-1 assay. The absorbance at 450 nm is shown. One-way ANOVA (n ≥ 3 independent experiments) * *p* < 0.05; ** *p* < 0.01 vs. invasion of UT-SCC-15.

**Figure 4 cancers-14-02484-f004:**
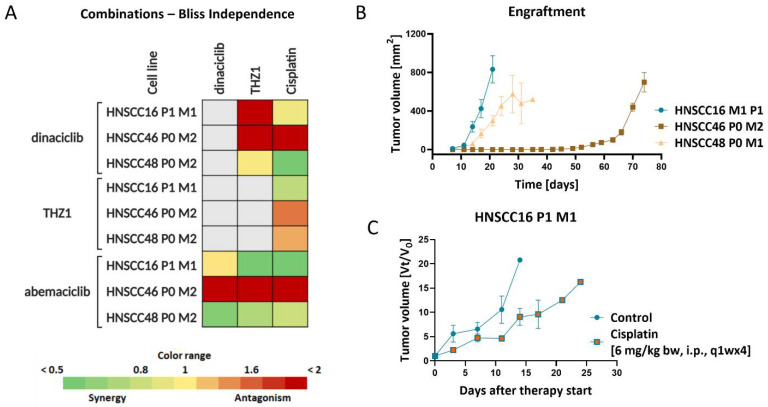
In vitro and in vivo drug response. (**A**) The Bliss Independence model was used to calculate potential synergistic or additive effects. The green color indicates a synergistic and red color indicates an antagonistic effect of the simultaneous combinations. Drug doses were applied for 2 × 72 h as follows: abemaciclib [HNSCC16 P1 M1/HNSCC46 P0 M2: 1 µM, HNSCC48 P0 M1: 0.1 µM], dinaciclib [HNSCC16 P1 M1/HNSCC46 P0 M2: 1 nM, HNSCC48 P0 M1: 5 nM], THZ1 [HNSCC16 P1 M1/HNSCC48 P0 M1: 20 nM, HNSCC46 P0 M2: 10 nM] and Cisplatin [HNSCC16 P1 M1/HNSCC46 P0 M2: 1 µg/mL, HNSCC48 P0 M1: 0.05 µg/mL]. (**B**) Engraftment of the cell lines HNSCC16 P1 M1 (green), HNSC46 P0 M2 (red), and HNSCC48 P0 M1 (blue) in NSG mice. (**C**) Therapy approach. Tumor growth curve of HNSCC16 P1 M1. Tumor volume was calculated as tumor volume on day × (Vt) divided through tumor volume at the therapy start (V0).

**Figure 5 cancers-14-02484-f005:**
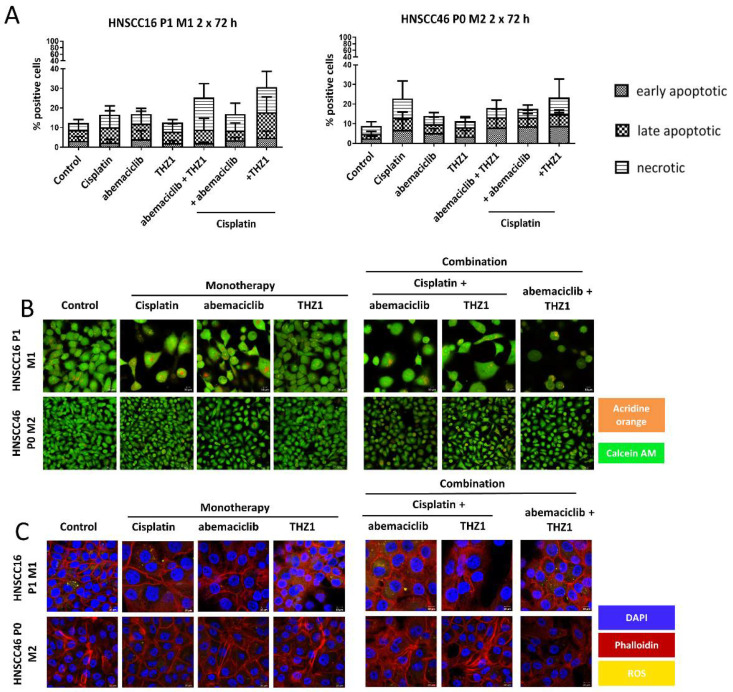
Apoptosis-necrosis assay, detection of acidic components, the cytoskeleton, and reactive oxygen species (ROS). Drug doses were abemaciclib [HNSCC16 P1 M1/HNSCC46 P0 M2: 1 µM], THZ1 [HNSCC16 P1 M1/HNSCC46 P0 M2: 10 nM], and Cisplatin [HNSCC16 P1 M1/HNSCC46 P0 M2: 1 µg/mL]. (**A**) Apoptosis-necrosis assay was studied after 2 × 72 h treatment. Cells were stained with Yo-Pro 1 iodide and PI. Early apoptotic cells were defined as positive for Yo-Pro 1 iodide, late apoptotic cells were defined as positive for Yo-Pro 1 iodide and PI and necrotic cells were defined as positive for PI. Kruskal–Wallis (n = 4 independent experiments) vs. control. (**B**) Semi-quantitative analysis of acidic components and viability of the 2 × 72 h treated cells was carried out with acridine orange and Calcein AM. Representative images are shown. A read out was conducted with the ZEISS Elyra 7 Confocal Laser Microscope (Zeiss). Original magnification 400×. (**C**) Semi-quantitative analysis of the cytoskeleton and ROS of the 2 × 72 h treated cells. A readout was conducted with the ZEISS Elyra 7 Confocal Laser Microscope (Zeiss). Original magnification 630×.

**Figure 6 cancers-14-02484-f006:**
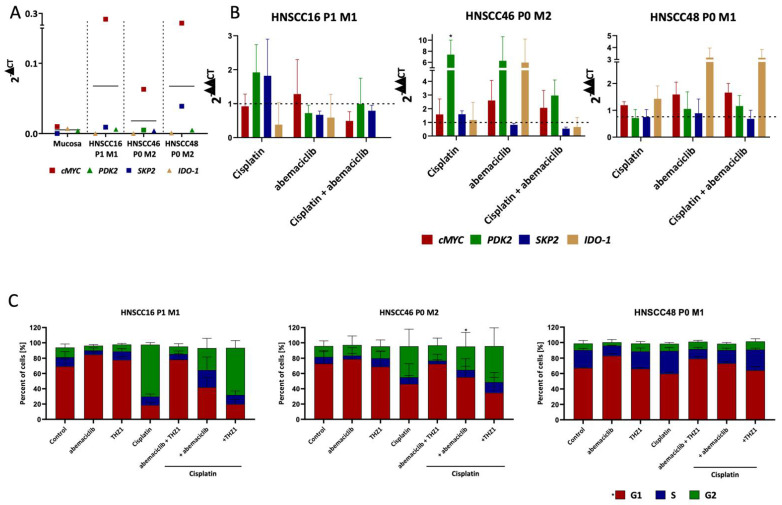
Expression of resistance-associated genes *(SKP2, cMYC, PDK2,* and *IDO-1)* and cell cycle analysis (**A**) Quantitative real-time PCR was carried out on untreated HNSCC cell lines and compared to benign mucosa (**B**) Influence of 2 × 72 h treatment on the expression of the resistance-associated genes *cMYC, PDK2, SKP2,* and *IDO-1*. Drug doses were abemaciclib [HNSCC16 P1 M1/HNSCC46 P0 M2: 1 µM, HNSCC48 P0 M1: 0.1 µM], and Cisplatin [HNSCC16 P1 M1/HNSCC46 P0 M2: 1 µg/mL, HNSCC48 P0 M1: 0.05 µg/mL]. Kruskal–Wallis (n = 3 independent experiments) * *p* < 0.05 vs. control. For *PDK2* and *SKP2* genes in HNSCC46 after THZ1- and Cisplatin treatment qPCR could only be performed two times because of poor RNA quality. (**C**) Cell cycle analysis was performed after 2 × 72 h of treatment. Drug doses were: abemaciclib [HNSCC16 P1 M1/HNSCC46 P0 M2: 1 µM, HNSCC48 P0 M1: 0.1 µM], THZ1 [HNSCC16 P1 M1/HNSCC48 P0 M1: 20 nM, HNSCC46 P0 M2: 10 nM], and Cisplatin [HNSCC16 P1 M1/HNSCC46 P0 M2: 1 µg/mL, HNSCC48 P0 M1: 0.05 µg/mL]. Cells were ethanol fixed and stained with PI. Quantitative analysis of the flow cytometry data was performed using using BD FlowJo software. Given are cells in the G1-, S- and G2-phase. Kruskal–Wallis (n ≥ 3 independent experiments) * *p* < 0.05 vs. control.

**Figure 7 cancers-14-02484-f007:**
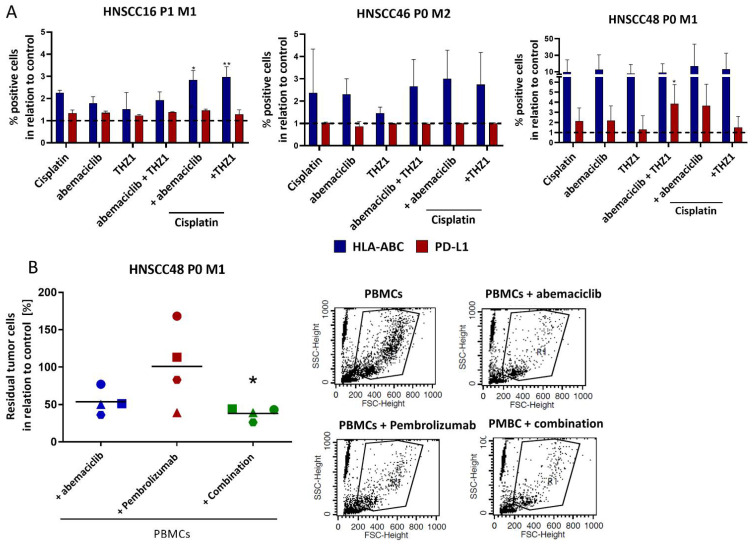
Phenotyping and allogenic co-culture. (**A**) Phenotyping was carried out after 2 × 72 h treatment. Drug doses were: abemaciclib [HNSCC16 P1 M1/HNSCC46 P0 M2: 1 µM, HNSCC48 P0 M1: 0.1 µM], THZ1 [HNSCC16 P1 M1/HNSCC48 P0 M1: 20 nM, HNSCC46 P0 M2: 10 nM], and Cisplatin [HNSCC16 P1 M1/HNSCC46 P0 M2: 1 µg/mL, HNSCC48 P0 M1: 0.05 µg/mL]. Cells were stained with anti-HLA-ABC antibody (MHC I) and anti-CD274 (PD-L1) after 2 × 72 h treatment. Shown are the HLA-ABC and PD-L1 positive cells are illustrated in relation to the control. Kruskal-Wallis (n = 3 independent experiments) * *p* < 0.05 ** *p* < 0.01 vs. control. (**B**) Allogenic co-culture. The cell line HNSCC48 P0 M1 was simultaneously treated for 1 × 72 h with (1) abemaciclib in combination with PBMCs (blue), (2) Pembrolizumab in combination with PBMCs (red), or (3) abemaciclib with Pembrolizumab and PBMCs (green). Drug doses were: abemaciclib 0,1 µM, Pembrolizumab 10 µg/mL. The ratio of target to effector cells was 1:10. The PBMCs were stimulated with 100 IU/mL IL-2. After 72 h, the residual tumor cells were counted by adding fluorescent beads. A readout was conducted with the FACS. The squares indicate for donor A, the triangle for donor B, the circles for donor C, and the hexagon for donor D. In the right part, representative FACS dot plots of treated cells with PBMCs and PBMCs in combination with abemaciclib are shown. One-way ANOVA (n = 4 independent experiments) * *p* < 0.05 vs. control (100%).

**Table 1 cancers-14-02484-t001:** Clinical characteristics.

Cell Line	Age Sex	TNM Grade	Localization	Origin	p16/HPV	Noxes	Treatment	Clinical Follow-Up
HNSCC16 P1 M1	82 m	rpT4a pN0 cM0 G2	Larynx	Recurrence	-/-	non-smoker	surgery 2008, radiation 2015, surgery 2020	alive
HNSCC46 P0 M2	69 m	pT3 pN3b cM0 G2	Hypopharynx	primary tumor	+/-	nicotine, C2	surgery (later adjuvant RCT)	†
HNSCC48 P0 M1	63 m	pT3 pN3b cM0 G2	Lymph node (primary tumor: oral cavity (floor of moth))	metastasis/ recurrence	-/-	non-smoker	RCT (64 Gy, Cisplatin), surgery (later Nivolumab)	†

RCT: Radiochemotherapy; +: positive; -: negative; †: dead

**Table 2 cancers-14-02484-t002:** Comparative analysis of mutational profile between patients’ tumor, the corresponding PDX and cell line.

Mutation	HNSCC16			HNSCC46			HNSCC48		
	Patient	PDX	Cell Line	Patient	PDX	Cell Line	Patient	PDX	Cell Line
*TP53*	p.R306 * VAF 37% p.P72R VAF 32.7%	p.R306 * VAF 100%	p.R306 * VAF 99.5%	p.E294Sfs *51 VAF 64.5% p.P72R VAF 100%	p.E294Sfs *51 VAF 99.3% p.P72R VAF 99.3%	p.E294Sfs *51 VAF 99.9% p.P72R VAF 99.6%	p.R175H VAF 99.4% p.P72R VAF 99.5%	p.R175H VAF 60.5% p.P72R VAF 80.4%	p.R175H VAF 99.8% p.P72R VAF 99.9%
*CDKN2A*	wt	wt	wt	wt	wt	wt	p.R58 * VAF 99.6%	p.R58 * VAF 55.8%	p.R58 * VAF 99.8%
*SMAD4*	wt	wt	wt	wt	wt	wt	p.R135 * VAF 99.6%	p.R135 * VAF 48.2%	p.R135 * VAF 100%

VAF: Variant allele frequency; Wt: Wild type; *: stop codon. The specific mutation loci and the VAF are shown. Analysis was carried out with the Illumina Cancer Hotspot Panel.

**Table 3 cancers-14-02484-t003:** IC_50_ values of test substances as determined by crystal violet staining.

Substance	HNSCC16 P1 M1	HNSCC46 P0 M2	HNSCC48 P0 M1
Cisplatin [µg/mL]	1.43	1.54	0.30
5-FU [µg/mL]	0.31	0.17	0.04
Cetuximab [µg/mL]	not reached	not reached	137.26
dinaciclib [nM]	2.69	1.65	7.33
THZ1 [nM]	49.22	39.47	32.00
abemaciclib [nM]	not reached	not reached	696.65

Cells were treated for 2 × 72 h with the different test substances in varying concentrations between 0.048 µg/mL–1 mg/mL for approved drugs (Cisplatin, 5-FU, and Cetuximab) and 1 nM–1 µM for targeted substances (dinaciclib, abemaciclib, and THZ1). Based on the dose–response curves IC_50_ was calculated using the IC_50_ calculator from AAT Bioquest (https://www.aatbio.com/tools/ic50-calculator/ (accessed on 1 November 2021)). If the highest dose applied did not result in a reduction in biomass compared to the control, this was defined as IC_50_ not reached.

## Data Availability

The data presented in this study are available in this article (and Appendix A).

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
