# Peer review of "Preclinical Head and Neck Squamous Cell Carcinoma Models for Combined Targeted Therapy Approaches"

_cancers, 2022, doi:10.3390/cancers14102484_

Round 1

Reviewer 1 Report

The work raises a very important topic. The results of HNSCC treatment
are unsatisfactory. It is very important to find new therapies
to improve treatment outcomes.
The test methods have been described and illustrated in great detail.
A very current and interesting topic.
I recommend that you publish the article without corrections.

Author Response

Dear Reviewer,

thank you very much for the excellent review of our manuscript " Preclinical head and neck squamous cell carcinoma models for combined targeted therapy approaches " by Schoenwaelder et al..

We thank you very much for the recommendation to publish our manuscript without any further changes.

In the name of all authors,

Yours sincerely,

Claudia Maletzki

Reviewer 2 Report

The study want to show the role of xenograft-derived HNSCC cell line models to guide selection of drug combinations and to detect resistance mechanisms. A relevant limitation of the work is the low number of HNSCC cell lines involved (n=3) to address this crucial aim. 

To that end, authors should clarify certain confusions in the interpretation of their data and add some data to support their conclusions. 

  • In the text authors indicated that all cell lines were HLA-ABC+ and PD-L1+, but the information is not fully in agreement with Fig. 3A. In the printed version of the paper Fig. 3A is also too small to simply appreciate data.
  • In Fig. 3C representative photo of migration of HNSCC48 P0 M1 at day 1 should be included to make more effective the comparison.
  • Table 3 is not sufficient to describe drug response analysis. Authors should include cytotoxicity assays (e.g. cytofluorimetric analysis or bioluminescence assays) performed at IC50 for each drug and drug combinations involved in the study, as included in Fig. 5-6-7. 
  • In Fig. 4C is shown in vivo activity of Cisplatin on HNSCC16 M1 P1. Why authors did not test in vivo other drugs and/or drugs combinations involved in the study? 
  • It is not clear the time of treatment used for drug testing (e.g. 2x72 h, 1x72 h), both in matherial&methods and results. 

Author Response

Dear Reviewer,

thank you very much for the excellent review of our manuscript " Preclinical head and neck squamous cell carcinoma models for combined targeted therapy approaches " by Schoenwaelder et al.. We appreciate the helpful comments and modified the text according to your suggestions. Modifications in the text of the manuscript are marked up using the “Track Changes” function of MS Word.

We would like to answer on a point-to-point basis:

  • The reviewer noticed some disagreement between the text and the corresponding Figure 3A. We thank the reviewer for critical reading and apologize for mislabeling the x-axis. We corrected the text and the figure accordingly.
  • Following the same line, the reviewer pleased us to increase the size of the illustrations depicted in Figure 3A. This has been done accordingly. Please find the updated Figure 3 embedded in the main text on page 9.
  • Another point raised by the reviewer is again in Figure 3. Here, the reviewer wished incorporation of a representative photo of migration of HNSCC48 P0 M1 at day 1. This has been done accordingly.
  • Then, the reviewer complained about the experimental setting by using the IC20-IC30 for drug response analysis and suggested using IC50 for each drug and drug combinations involved in the study. We thank the reviewer for this advice, however, we respectfully disagree. First, drug response analysis was done by using varying concentrations of each individual test substance (please see also material and methods section 2.15 for further details). Read out was done with crystal violet staining. This is a classical method to determine viability of adherent cells via biomass quantification. We established this method in our lab a couple of years ago and compared the results with different other methods (e.g. cytofluorimetric Calcein AM staining, flow cytometric CMFDA staining). Finally, we decided to use crystal violet for initial sensitivity screening, followed by more advanced and sophisticated methods on selected drugs with high cytotoxic activity. With regard to the suggested IC50 we would like to explain the rationale for not using such a high dose for the combination approach as follows: (I) the IC50 is a relatively high dose and in some cases, it even exceeds the plasma concentration reachable in vivo; (II) some drugs may enhance side effects when used in combination, therefore, the dose should be minimized; (III) we aimed to identify synergistic effects, this is hardly achievable when IC50 values are used for each substance.
  • The next point addressed by the reviewer is about the treatment schedule used for in vivo experiments and the rationale for choosing Cisplatin instead of other drugs. We totally agree with the reviewer that testing other drugs would be very interesting. However, for this initial trial we focused on Cisplatin as clinically approved standard drug for the treatment of HNSCC patients both in mono- or combination therapy. Also, we wanted to get an idea on the sensitivity of our novel tumor models in vivo. For ethical reasons, we omitted application of other therapeutic agents here and focused on in vitro analyses.
  • Finally, the reviewer wished the time of treatment used for drug testing (e.g. 2x72 h, 1x72 h), both in material & methods and results to be more clearly mentioned in the manuscript. This has been done accordingly. 

We are very confident that the revised version now matches the requirements for publication in Cancers. We would be very pleased if you would find this enhanced version suitable for publication.

In the name of all authors,

Yours sincerely,

Claudia Maletzki

Round 2

Reviewer 2 Report

Dear authors,

thank you very much for your adjustments of the manuscript according to my suggestions.

I would like to ask the authors to clarify some points yet:

  • For the point of in vitro drug response, I agree with your explanations about crystal violet staining read out and your concerns on using IC50 dose for in vitro drug testing. However, my suggestion is not linked to the usage of IC50 dose per se but to the fact that are no shown data of crystal violet staining or relative biomass quantifications, except for table 3. To my mind, table 3 is not immediate and effective to describe in vitro drug testing (as outlined in Material&Methods section 2.13). A more detailed figure panel is required as done for Fig. 5 for example.
  • I saw the time of treatment mentioned in the appropriate sections of the text but what does it means 2x72h or 1x72h, technically? It is still not clear to me.
